# Recent Advances in Models, Mechanisms, Biomarkers, and Interventions in Cisplatin-Induced Acute Kidney Injury

**DOI:** 10.3390/ijms20123011

**Published:** 2019-06-20

**Authors:** Sara J. Holditch, Carolyn N. Brown, Andrew M. Lombardi, Khoa N. Nguyen, Charles L. Edelstein

**Affiliations:** Division of Renal Diseases and Hypertension, University of Colorado at Denver, Box C281, 12700 East, 19th Ave, Aurora, CO 80045, USA; sara.holditch@ucdenver.edu (S.J.H.); CAROLYN.N.BROWN@UCDENVER.EDU (C.N.B.); anlo5057@colorado.edu (A.M.L.); KHOA.N.NGUYEN@UCDENVER.EDU (K.N.N.)

**Keywords:** cisplatin, acute kidney injury, AKI, apoptosis, inflammation, oxidative stress

## Abstract

Cisplatin is a widely used chemotherapeutic agent used to treat solid tumours, such as ovarian, head and neck, and testicular germ cell. A known complication of cisplatin administration is acute kidney injury (AKI). The development of effective tumour interventions with reduced nephrotoxicity relies heavily on understanding the molecular pathophysiology of cisplatin-induced AKI. Rodent models have provided mechanistic insight into the pathophysiology of cisplatin-induced AKI. In the subsequent review, we provide a detailed discussion of recent advances in the cisplatin-induced AKI phenotype, principal mechanistic findings of injury and therapy, and pre-clinical use of AKI rodent models. Cisplatin-induced AKI murine models faithfully develop gross manifestations of clinical AKI such as decreased kidney function, increased expression of tubular injury biomarkers, and tubular injury evident by histology. Pathways involved in AKI include apoptosis, necrosis, inflammation, and increased oxidative stress, ultimately providing a translational platform for testing the therapeutic efficacy of potential interventions. This review provides a discussion of the foundation laid by cisplatin-induced AKI rodent models for our current understanding of AKI molecular pathophysiology.

## 1. Introduction

Cis-diamminedichloroplatinum(II) (cisplatin) is an inorganic platinum derivative used to treat several types of cancer including bladder, ovarian, lung, and testicular [1]. Cisplatin is prescribed to 10–20% of all cancer patients due to its efficacy in slowing cancer growth. Renal uptake and excretion of cisplatin is mediated by proximal tubule localized transporters, such as OCT2 and MATE1. Consequently, cisplatin accumulates in renal proximal tubular cells, resulting in inflammation, injury, and cell death [2].

Acute kidney injury (AKI) is the principal limitation for cisplatin cancer treatment, affecting 30% of cisplatin recipient patients. AKI is clinically defined by a decline in kidney function that manifests as a ≥0.3 mg/dL increase in serum creatinine or a 0.5 mL/kg/h decrease in urine output [3]. Several additional biomarkers, such as NGAL, KIM-1, and Cystatin-C, are routinely used to detect AKI in both patients and animal models [4], but are not always reliable indicators of AKI, as they are less sensitive to mild cases of kidney injury [5] and present differently based on factors such as age [6], sex, and patient comorbidities.

Despite several positive treatment studies in rodents, the prevalence of cisplatin-induced AKI in patients remains high, for several possible reasons. The goal of cisplatin dosing regimens in patients is to slow cancer growth while limiting off-target side effects, such as nephrotoxicity. Rodents are purposely given large doses of cisplatin with the intent of causing severe and easily detectable AKI (i.e., significant increase of current known biomarkers). Thus, there is a discrepancy between the manifestation of AKI in humans versus mouse models. Therefore, it is crucial to identify novel biomarkers of AKI, refine animal models, and identify new molecular targets for treatment. Recently, the field has expanded on the types of cisplatin-induced AKI models utilized, the pathomechanism of cisplatin-induced injury unique to reactive oxygen species and mitochondrial dysfunction, pathways of cell death, inflammatory responses, autophagy, and the role of Klotho in AKI [7,8,9,10,11,12,13,14,15,16,17,18,19].

### 1.1. Cellular Import and Export Mechanisms

Cellular uptake of cisplatin is mediated by basolaterally-localized influx transporters such as organic cation transporters (OCTs) and copper transporters (CTRs) [20]. Cisplatin secretion into the urine is mediated by apically-localized efflux transporters, such as multidrug resistance-associated proteins (MRPs), multi-antimicrobial extrusion protein (MATEs) [21], and ATPases [20,22]. These transporters are highly expressed in the proximal and distal tubules of the kidneys [23]. Aberrant expression and localization of cisplatin transporters in response to cisplatin administration has been reported in recent studies of cisplatin-AKI [21]. Inhibition of cisplatin influx transporters or activation of cisplatin efflux transporters have been popular strategies for reducing cisplatin-induced AKI. For example, cimetidine has been postulated to protect against nephrotoxicity by competitively inhibiting cisplatin transport via OCT2 in kidney cells, however this theory has been discredited in in vitro experiments in Madin-Darby Canine Kidney (MDCK) cells [24].

It should be noted, however, that inhibition of transporters that are also expressed in cancer cells may decrease the anti-cancer properties of cisplatin. Thus, it is important to use tumour bearing mouse models to test the renoprotective effects of therapies targeting cisplatin transporters.

Magnesium is a key regulator of expression of cisplatin transporters [25]. Cisplatin decreases serum magnesium [26]. Magnesium deficiency increases AKI severity [27] by decreasing expression of efflux transporters and allowing for cisplatin accumulation in tubular cells [28,29]. Kumar et al. recently showed that magnesium supplementation protected against AKI and did not interfere with the anti-tumour effects of cisplatin [27].

In conclusion, administration of cisplatin is shown to change expression of cisplatin transporters. Decreasing expression of influx transporters and increasing expression of efflux transporters is shown to protect against cisplatin-induced AKI. Targeting cisplatin transporters presents challenges, such as maintenance of the anti-tumour efficacy of cisplatin. However, targeting the renal uptake and excretion is an attractive therapeutic target as it allows for intervention before development of tubular injury by dysregulated intracellular signaling.

### 1.2. Cisplatin-Induced AKI Mouse Models

#### 1.2.1. Non-Tumour Bearing

Current research in cisplatin-induced AKI utilizes primarily two murine models, the short-term high dose [2], and long-term low dose nephrotoxic mouse models [30]. For example, the long-term model uses 5–15 mg/kg cisplatin administration, administered 2–4 times, over a period of 3–4 weeks [30]. The short-term model uses a single high dose of cisplatin, 20–30 mg/kg, that results in mortality and nephrotoxicity at 3–7 days post-cisplatin-induced AKI [2]. Many of these models rely heavily on male, young, 6–10 week, C57BL/6 mice. However, female C57BL/6 mice have a documented higher propensity for increased AKI severity after cisplatin injection [5]. Further, aged mice, 16–17 months of age, also demonstrate worse cisplatin-induced AKI compared to younger controls [6]. Even strain variance has influenced the development and severity of AKI [31,32]. Irrespective of the model, cisplatin-induced AKI is routinely measured using serum creatinine, BUN, Cystatin-C, KIM-1, NGAL, CCL2, and IL-18 (Table 1). Interestingly, novel biomarkers, such as renal specific miRNAs, have been described in cisplatin-induced AKI (Table 1). MiRNAs are useful in detecting cisplatin-induced AKI because they show increases in the urine after 18 h, which is faster than creatinine or BUN [4]. Biomarkers may provide utility in detecting AKI with high sensitivity and specificity, prior to decreased kidney function.

#### 1.2.2. Tumour Bearing Models

The two most common cisplatin-induced AKI models discussed above are by and large the work horse of understanding the pathomechanism of cisplatin-induced inflammation, apoptosis, reactive oxygen species, and cell populations mediating inflammation. Though informative, neither of these two models reflect the clinical reality for patients [53]. For example, cisplatin is administered chronically, at doses less than 10 mg/kg in patients [30], in order to treat solid tumours. Yet, rarely are mice consistently administered less than 10 mg/kg of cisplatin, nor do studies incorporate tumour bearing mice. While the critique on dosing simplifies the technical reality that certain strains of mice may be more resistant to nephrotoxic substances [54], necessitating non-clinically relevant doses to produce clinically relevant cisplatin AKI phenotypes, it highlights the room for improvement on clinically relevant small animal models and the inherent difficulty in translating observations made from the bench to the bedside.

Further, cisplatin is known to cause AKI in approximately 1/3 of patients [55]. However, perimenopausal women have a significantly higher incidence of developing cisplatin-induced AKI compared to men of the same age [56], again, not reflected in the current animal models either in age or sex. To date, very few preclinical and clinical trials have been performed to discover preventative treatments for chemotherapy- induced AKI in cancer patients. As such, it stands to reason that using more applicable animal models (i.e., tumour bearing aged male and female mice receiving physiological dosing schedules) will greatly improve the translatability and reproducibility of the models. It is important to perform preventative or interventional studies addressing the pathophysiology of cisplatin-induced AKI and accounting for sex and in vivo tumours.

Allograft models are the bulk of tumour bearing cisplatin-induced AKI models, and offer the advantage of using immunocompetent hosts, short latency of engraftment, but are limited by a shortage of useful mouse cell lines [57]. Cell lines used for AKI allograft models include: murine derived EL4 lymphoblastic cells [58], H22 hepatocellular carcinoma [59], CMT167 pulmonary adenocarcinoma [50,60], and CT26/WT fibroblast colon carcinoma [27]. In the rapid cisplatin-induced AKI model, allografts are given 7–10 days to form solid tumours. Once tumours are formed, mice are given single doses of cisplatin (20–25 mg/kg) with or without nephroprotective therapies like PLA2 [61], Suramin [41], recombinant human MG53 [62], or G31P, a CXCR1/2 antagonist [59].

Long-term models allow tumour engraftment as seen in rapid cisplatin-induced AKI models, followed by cisplatin treatment (3.33 mg/kg-10 mg/kg) every 3–7 days for 3–4 weeks. These studies aim to address the chemotherapeutic value of secondary variables like CD4 knockout [60], Magnesium supplementation [27,29], IL-33 deficiency [50], and Mangiferin supplementation [40] on I) potentially synergistic effects on tumour intervention and II) nephroprotection.

In summary, while conventional short-term and long-term cisplatin-induced AKI murine models have produced a profound amount of insight into the pathobiology and molecular mechanisms of cisplatin-induced AKI, there is room for improvement when it comes to administering physiological dosing regiments of cisplatin (<10 mg/kg, chronic vs. acute), incorporating age, gender, background strain, and tumour allografts.

### 1.3. Cell Stress

#### 1.3.1. Reactive Oxygen Species and Mitochondrial Dysfunction in Cisplatin-Induced AKI

Oxidative stress induced by mitochondrial dysfunction and accumulation of intracellular reactive oxygen species (ROS) is a hallmark of cisplatin-induced AKI. Several studies have shown that cisplatin administration is associated with impaired mitochondrial function, increased oxidative stress, and dysregulated expression of endogenous antioxidant enzymes. Targeting these processes by promoting mitochondrial biogenesis, enhancing mitochondrial dynamics, or increasing availability of endogenous antioxidants has demonstrated a protective effect by decreasing oxidative stress and downstream consequences such as cell death in cisplatin-induced AKI [63,64,65].

A substantial source of ROS production lies in the mitochondria, the organelle responsible for producing cellular ATP. Mitochondria require narrow parameters (e.g., maintenance of proton gradient across mitochondrial membrane, expression and function of complexes involved in the electron transport chain, transportation of metabolites into the mitochondria) in order to maintain function. Deviation from these parameters due to cellular stress results in mitochondrial uncoupling, increased ROS production, and decreased ATP production.

Cisplatin is known to directly and indirectly regulate mitochondrial function. Upon receptor-mediated endocytosis, cisplatin is hydrolyzed into a positively charged molecule. As an electrophile in the cytoplasm, cisplatin directly disrupts the action of mitochondrial complexes, leading to increased production of ROS [66]. Recent studies of cisplatin-induced AKI have focused on indirect effects of cisplatin on mitochondrial function. For example, cisplatin causes upregulation of miR-709, which is shown to interact with and inhibit mitochondrial transcription factors, such as mitochondrial transcription factor A (TFAM) [67]. Further, genetic disruption of transcription factors regulating mitochondrial biogenesis, such as the estrogen receptor alpha (ERα), has been shown to exacerbate cisplatin-induced AKI [68]. Decreased transcription of mitochondrial genes ultimately leads to decreased function of mitochondria (lower oxygen consumption rate, increased superoxide production) and increased apoptosis [67]. Conversely, use of an antagomiR against miR-709, stabilizes TFAM, enhancing transcription of mitochondrial genes and resulting in protection against AKI [67]. In addition to mitochondrial biogenesis and function, maintenance of mitochondrial dynamics (i.e., fusion, fission) is critical in cellular response to stressors (i.e., creating new mitochondria to meet energetic demands, removing damaged mitochondria) [69]. Mitochondrial localized (NAD)-dependent deacetylases, such as sirtuins (SIRT), indirectly regulate mitochondrial dynamics through modification (i.e., deacetylation) of other mitochondrial proteins. Promoting mitochondrial fusion and dynamics using SIRT activators (specifically, SIRT3) protects against AKI [65,70,71,72]. In summary, loss of mitochondrial function and integrity by decreased transcription of mitochondrial genes or decreased SIRT-dependent regulation of mitochondrial dynamics leads to increased ROS production, decreased ATP production, and negative downstream consequences including cell death.

Heathy mitochondria produce ROS as a normal product of cellular metabolism as well as in response to signaling molecules such as cytokines [73]. Under cellular stresses or stimuli, such as cisplatin administration, mitochondria significantly increase their production of ROS. Further, cisplatin increases intracellular accumulation of ROS by decreasing expression of endogenous antioxidant enzymes such as superoxide dismutase (SOD), glutathione (GSH), and catalase (CAT) [49,64,65,74,75,76,77,78,79,80,81,82,83,84,85,86,87,88,89]. Oxidative stress occurs when the cell is unable to respond to, or neutralize, the amount of ROS being generated. If not properly neutralized, ROS cause dysregulation of several signaling pathways including MAPK, PI3K, Nrf2, iron metabolism, DNA damage response, and cell death [73]. Thus, enhancing the cellular antioxidant response to combat increased ROS has been a common therapeutic approach in cisplatin-induced AKI. Several novel antioxidant compounds have been shown to protect against cisplatin-induced AKI by increasing expression of endogenous antioxidant enzymes such as SOD, GSH, and CAT [65,74,76,81,82,89,90]. In order for antioxidants to exert their full function, it is important that they cross the plasma membrane and localize to the site of ROS generation, the mitochondria [91]. Thus, targeting of antioxidants to the mitochondria has been used as a novel antioxidant therapeutic intervention in cisplatin-induced AKI [49].

The main downstream consequence of uncontrolled ROS accumulation is cell death. However, the role of ROS in programmed versus accidental cell death is unclear [92]. Specifically, ROS have been shown to activate death receptor, mitochondrial, and ER-mediated apoptosis pathways [93]. Further, AKI is exacerbated by increased necroptosis in mice overexpressing NADPH Oxidase 4 (NOX4) through increased production of ROS [94]. Unchecked production of ROS is a major contributor to tubular injury in cisplatin-induced AKI, making antioxidant-based therapies a popular choice for therapeutic intervention in mouse models. Loss of mitochondrial integrity is a key contributing factor to excessive production of ROS, decreased ATP production, and subsequent amplification of cellular stress and death. Targeting these processes prevents adverse downstream effects, such as uncontrolled cell death and tubular injury.

#### 1.3.2. Markers of Oxidative Stress and Mitochondrial Dysfunction

The most commonly used markers of oxidative stress in cisplatin-induced AKI are malondialdehyde (MDA), NADPH oxidases (NOX), and heme oxygenase 1 (HO-1). Further, the most commonly measured endogenous antioxidant enzymes are SOD, GSH, and CAT. Activity of these enzymes can be measured easily in vitro or in vivo using commercially available activity assays. Total expression of endogenous antioxidant enzymes can be measured using immuno-based assays, such as immunoblotting, immunofluorescence, immunohistochemistry, and ELISAs.

Further, two examples of cellular superoxide indicators include the cell-permeable dye, dihydroethidium (DHE), and MitoSOX. DHE fluorescence shifts when it is oxidized and MitoSOX fluoresces when oxidized by superoxide within the cell [95]. In addition to superoxide indicators, mitochondrial dyes and indicators have been developed in order to measure mitochondrial function, size, and shape. MitoTracker, for example, stains mitochondria, regardless of function. MitoSensor, however, detects the proton gradient across the mitochondrial membrane, making it an excellent indicator of mitochondrial function. Unfortunately, most of these techniques require the use of live cells, limiting their application for in vivo studies.

### 1.4. Cell Death

#### 1.4.1. Apoptosis in Cisplatin-Induced AKI

Although cisplatin-induced tubular cell death is generally a secondary consequence of upstream cellular dysfunction, such as oxidative stress, it is a critical point of therapeutic intervention. Excessive cell death through necrosis can trigger inflammatory processes, which may propagate tubular injury [96]. Histological scoring of tubular cell death (acute tubular necrosis or ATN) is a common measure of AKI severity in patients and animal models of cisplatin-induced AKI. Upstream of cell death lie apoptosis and necrosis, two distinct pathways at the nexus of cisplatin-induced DNA damage, inflammation, and oxidative stress [97].

Apoptosis is primarily mediated by cysteine-aspartic proteases (caspases), produced as inactive proteins that must be dimerized, cleaved, or both, to be activated. In short, an extracellular or intracellular stimulus causes cleavage of an initiator caspase (i.e., caspase 8, caspase 9), which then cleaves an executioner caspase (i.e., caspase 3, caspase 7), resulting in DNA fragmentation, nuclear condensation, and cell death. Though several initiator caspases exist, the main executioner caspase implicated and measured in cisplatin-induced AKI is caspase 3. The degree to which caspase 3 activates apoptosis is dependent on the dose of cisplatin [98]. Cleavage of caspase 3 occurs in response to activation of one or more of the three apoptotic pathways: I) extrinsic (death receptors), II) intrinsic (mitochondrial), or III) ER (endoplasmic reticulum) mediated.

The extrinsic pathway of apoptosis is mediated by intracellular caspase activation through death receptor activation by FasL, TRAIL, or TNFα. Fas ligand (FasL) is expressed on the surface of immune cells, such as lymphocytes, and binds the Fas receptor on target cells. FasL is upregulated in the kidney in response to cisplatin and can be reduced by treatment with herbal medicine, such as Hydrangea paniculata [99]. TNFα binding and activation of the TNFα receptor is an extremely prevalent apoptosis stimulus in models of cisplatin-induced AKI. Interestingly, TNFα can be pro-apoptotic and pro-survival simultaneously through caspases and NFκB pathways, respectively [100]. TNFα is produced mainly by macrophages [101], and binds TNFR, initiating an intracellular caspase cascade that ends in death. A more detailed discussion of the mechanism and interventional treatments are discussed below. The main initiator caspase that is cleaved in response to activation of the above mentioned receptors is caspase 8. Kidney cleaved caspase 8 is increased in response to cisplatin administration and is reduced by the FDA approved drug, suramin [41]. Increased cleavage of caspase 8 exacerbates AKI through increased cell death [102].

The intrinsic pathway of apoptosis is mediated in part by mitochondrial dysfunction. The tumour suppressor protein, p53, is activated in response to DNA damage. p53, though mainly known for its anti-cancer properties through increased cell death of cancer cells, is also implicated as being a key apoptotic stimulus in several models of AKI. p53 inhibits anti-apoptotic proteins localized to the mitochondrial membrane [103], such as the B-cell lymphoma (Bcl) family of anti-apoptotic proteins. Bcl family proteins are essential for maintenance of mitochondrial function and membrane integrity. There is a strong consensus that the anti-apoptotic mitochondrial protein, Bcl-2, is downregulated in response to cisplatin [40,44,70,75,76,78,79,83,84,88,89,104,105,106,107,108,109,110,111,112,113,114,115,116,117] through p53 activation [111,113,118]. One study, however, showed that Bcl-2 mRNA is increased after administration of cisplatin [119]. In the absence of these anti-apoptotic proteins, mitochondrial outer membrane permeabilization (MOMP) occurs [120]. MOMP allows for release of factors present in the mitochondria, such as cytochrome c, into the cytosol that propagate the caspase cascade through cleavage of caspase 9. Several herbal medicines, including Nelumbo nymphaea and QiShenYiQi Pills, protect against AKI by preserving mitochondrial function and decreasing cleavage of caspase 9, ultimately protecting against mitochondrial cell death and tubular damage [121,122]. Further, morin (flavanol) hydrate is shown to directly inhibit p53 activation and protect against tubular cell death in AKI [123]. In conclusion, several naturally-occurring and FDA-approved compounds exert protective effects in the kidneys of mice receiving cisplatin, primarily through their anti-apoptotic actions. Compounds with anti-apoptotic properties may potentially allow increased cancer growth due to decreased apoptosis in the cancer. This possibility highlights the need to test anti-apoptotic compounds in cisplatin-induced AKI in mice that have cancer to determine the effect of the compound on the cancer as well as the AKI.

ER stress induced apoptosis, though the least common cell death mechanism, is an important mediator of cisplatin-induced AKI. There are three main ER stress sensors, including inositol-requiring protein 1 (IRE1), protein kinase R (PKR)-like endoplasmic reticulum kinase (PERK), and activating transcription factor 6 (ATF6). Under normal conditions, these sensors are inactive, and bind ER chaperones BiP/GRP78 and XBP. However, with cisplatin administration, these markers are released, activated, and increase in abundance [124]. The mechanisms through which dysregulation of ER stress sensors contributes to AKI is not known. Increased polyamine catabolism is a hallmark of ER stress. Cisplatin administration is associated with an increase in the expression of enzymes involved in polyamine catabolism, such as SMOX and SSAT. Genetic disruption of these two enzymes protects against ER stress-dependent apoptosis in cisplatin-induced AKI [125]. Further, expression of ER oxidoreductase enzymes, such as ER oxidoreductin-1α (Ero1α), can be induced with hyperhomocysteinemia (HHcy) in vitro and in vivo [126]. HHcy in the absence of cisplatin-induced AKI increases Ero1α activity, producing excess hydrogen peroxide (H_2_O_2_), activates the unfolded protein response [127], and causes endothelial inflammation [126]. In the presence of cisplatin, mice with HHcy develop worse AKI severity, elevated ER stress, and severe renal tubular damage, relative to mice without abnormally high levels of homocysteine [128].

In addition to protein-protein interactions mediating the caspase cascade and resulting in cell death, pro- and anti-apoptotic proteins are regulated by cisplatin at the transcriptional level. For example, one study identified dysregulated methylation of apoptotic genes, such as interferon regulatory factor 8 (Irf8), after cisplatin administration [129]. Further, cisplatin upregulates expression of histone deacetylase (HDAC) 2. HDAC2 binds the promoter for BMP-7, an anti-apoptotic molecule. Class I and II HDAC inhibitors (TSA, VPA) restore BMP-7 expression and reduce cisplatin-induced apoptosis [110]. Additionally, some microRNAs (miRs) are shown to protect against AKI through cell death dependent mechanisms. For example, knockout of miR-155 exacerbates cisplatin-induced AKI by increasing activity of c-Fos, a gene that has been implicated in p53-mediated apoptotic cell death, independent of its activity as a transcription factor [130,131]. Further, NF-κB transcriptional inhibition in cisplatin-induced AKI ameliorates kidney function and reduces ATN scores without mediating a significant effect on apoptosis. These alterations are associated with a decrease pro-inflammatory mediators and caspase recruitment domain family, member 11 (CARD11) [119].

Based on the prominence of apoptosis induction in the setting of cisplatin-induced AKI, it would be expected that apoptosis inhibition would protect against cisplatin-induced AKI. To this point, erythropoietin is a known to have anti-apoptotic effects [132]. Multiple studies have demonstrated that administration of erythropoietin protects against cisplatin-induced AKI in both rats and mice associated with decreased tubular cell apoptosis [133,134]. Surprisingly, however, the effects of direct apoptosis inhibition, with caspase inhibitors, such as the pan-caspase inhibitor zVAD-fmk, resultes in worse AKI and worse ATN and apoptosis scoring in a mouse model of cisplatin–induced AKI [135]. Authors hypothesized that the pan-caspase inhibitor, zVAD-fmk, impaired autophagic flux by blocking autophagosome clearance, as evident by accumulation of autophagic substrates p62 and LC3-II. Chloroquine, a lysosomotropic agent known to impair autophagic flux, also exacerbates cisplatin-induced decline in renal function. These findings demonstrate that impaired autophagic flux, induced either by the pan-caspase inhibition or a direct autophagy inhibitor, will result in worsened renal function in cisplatin-induced AKI [135].

In summary, there are three apoptotic pathways carried out by a tightly coordinated intracellular proteolytic cascade. Reducing apoptosis is currently being targeted by post-translationally and transcriptionally based therapeutics. However, interventions in cisplatin-induced AKI that inhibit apoptosis should be studied in relation to autophagy as cell death is an important mediator of tubular injury.

#### 1.4.2. Necrosis and Necroptosis in Cisplatin-Induced AKI

While apoptotic cells die stealthily, necrotic cells do not undergo a coordinated intracellular program, and rather rupture and release their intracellular components. Necrosis, a form of non-programmed or “accidental” cell death, occurs in mouse models of AKI alongside apoptosis. Additionally, a new apoptotic-necrotic hybrid pathway, called necroptosis, has gained attention in cisplatin-induced AKI. Necroptosis is considered to be programmed, rather than accidental, and is mediated by coordinated activation of receptor interacting protein kinases (RIPKs) and mixed lineage kinase domain like pseudokinase (MLKL), ultimately resulting in permeabilization of the cell membrane. In addition to their role in apoptosis, caspases can also play a role in necroptosis through interaction with RIPKs. RIPK1 and 3 [94,111,136,137], pMLKL, and cleaved caspase 8 are upregulated in response to cisplatin administration [138]. Knockout of MLKL or RIPKs decreases necroptosis and protects against tubular damage [137]. A novel herbal compound, wogonin, protects against AKI, possibly through a physical interaction with the ATP-binding pocket of RIPK1 to inhibit its pro-necrotic effects [138]. Sustained necrosis after cisplatin administration is proposed to mediate the transition from AKI to chronic kidney disease (CKD) [139], making necrosis an important point for therapeutic intervention.

#### 1.4.3. Methods for Detection of Cell Death and Caveats

Protein expression of cleaved caspases is one of the most common markers of apoptosis in models of AKI. In addition to detection of specific caspases and other proteins, there are some non-specific methods for detecting cell death that are commonly used in rodent models of cisplatin-induced AKI. A commonly used marker to measure cell death is terminal deoxynucleotidyl transferase dUTP nick end labeling (TUNEL) staining, which detects the 3′ hydroxyl ends of fragmented DNA [140]. However, the TUNEL stain cannot differentiate between necrotic and apoptotic cell death, as both result in DNA fragmentation [141,142]. Necrosis is characterized by random DNA breakage/smear on agarose gel; apoptosis is characterized by non-random fragmentation/ladder on agarose gel; regardless, TUNEL detects any form of DNA fragmentation. Similar limitations are true for AnnexinV. AnnexinV binds phosphatidylserine (PS), which is flopped from the inner to the outer leaflet of the plasma membrane upon apoptotic cell death. Because necrosis causes rupture of the plasma membrane, the localization of PS cannot be detected. Thus, close histological examination, or measurement of proteins that are specific to each pathway, is required to confirm the mechanism by which cisplatin stimulates cell death. Lastly, the mechanism of cisplatin-induced cell death in vitro is dose-dependent, with lower doses inducing apoptosis and higher doses inducing necrosis [143,144]. In contrast, in vivo studies exhibit both types of cell death. Because cell death is such a prevalent phenomenon in cisplatin-induced AKI, the methods used to detect and define specific cell death mechanisms require refinement.

In conclusion, cell death results from a variety of stimuli including extracellular (activation of death receptors by cytokines) and intracellular (ROS mediated mitochondrial damage), and is seen in virtually all models of cisplatin-induced AKI. Treatments that target cell death pathways show promise in cisplatin-induced AKI. However, further studies are warranted to clarify the mechanisms of each distinct pathway and how they relate to tubular injury.

### 1.5. Inflammation in Cisplatin-Induced AKI

Inflammation is a necessary and evolutionary response developed in order to eliminate pathogens and mediate repair after injury. However, excess and prolonged inflammation can be injurious. For example, inflammation has a key role in promoting autoimmune disorders, fibrosis, and tissue damage. The pro-inflammatory nature of cisplatin has a well-documented detrimental role in acute kidney injury. Cytokines (interleukins, chemokines) released by leukocytes and injured renal tubular cells are instrumental in initiating and prolonging the extent of inflammation. Key cytokines involved in cisplatin-induced AKI can be classified by those that are I) autocrine or paracrine in nature, i.e., stimulating local and systemic inflammation, II) endocrine in nature, serving as biomarkers of AKI, and III) protective autocrine/paracrine cytokines. The following details the mechanistic role of cytokines within these subdivisions as they relate to murine models of cisplatin-induced AKI.

#### 1.5.1. Cytokines

Cisplatin increases both serum and urine concentrations of tumour necrosis factor alpha (TNFα), a pleiotropic cytokine with endocrine, paracrine, and autocrine pro-inflammatory consequences. In response to IL-1β, NF-κB, Sir1, and Deptor stimulation, TNFα can be produced by injured renal tubules [145], fibroblasts, keratinocytes, macrophages, and leukocytes [146]. For example, IL-1 receptor knockout (*IL-1R1*^−/−^) mice exhibit attenuated cisplatin-induced AKI and diminished levels of whole kidney TNFα [147]. Similarly, disrupted NF-κB signaling in Epoxide Hydrolase 2 knockout, *Ephx2*^−/−^ mice, attenuated NF-κB mediated transcription of TNFα [148], TNFR1, and TNFR2, ultimately reducing cisplatin-induced kidney injury. Likewise, deficiency of mTOR kinase interacting protein, DEPTOR, in the proximal tubules of cisplatin-treated mice ameliorated injury, inhibiting p38 activity and TNFα production [149]. Lastly, sirtuins (SIRTs), in addition to the role in ROS and mitochondrial function, are increasing in recognition for their importance as upstream stimulators and inhibitors of TNFα through their activation or inhibition of NF-κB. For example, kidney specific overexpression of SIRT1 [150] and SIRT6 [151] inhibit NF-κB activity, in turn suppressing expression of TNFα, leading to attenuated cisplatin-induced AKI [150]. Further, loss of SIRT7 expression through genetic disruption (*Sirt7*^−/−^ mice), decreases NF-kB activity, suppressing TNFα expression, and protecting against cisplatin-induced AKI [46].

Once produced and released extracellularly, TNFα can bind two receptors, TNF receptor 1 (TNFR1) or receptor 2 (TNFR2). TNFR1 is nearly ubiquitous among cell type expression, and upon binding TNFα, three pathways can be initiated; I) activation of NF-kB, II) activation of MAPK (JNK, p38-MAPK, and ERK), or III) induction of cell death signaling, such as Fas and caspase 8 induction (discussed above), ultimately leading to apoptosis or necrosis. TNFα stimulation can also induce systemic inflammatory responses, such as CCL2 (MCP-1), and CCL5 (RANTES). TNFR2 is constitutively expressed on CD4^+^Foxp3^+^ cells (Tregs), and is critical for activation, expansion, and functional stability of Tregs. Recently, TNFα exposure has been shown to induce Apo-A4 expression, a novel predictor for kidney injury via TNFR2 signaling. Removing TNFα signaling during cisplatin exposure, either by genetic disruption of upstream activators as described above, or genetic disruption of TNFα and TNF receptors specifically reduces the development of AKI [152,153]. For example, TNFα deficiency specifically in the kidney of chimeric mice treated with cisplatin [154], results in reduced renal dysfunction and renal injury. Indirectly reducing TNFα expression has also demonstrated protection against cisplatin-induced AKI. For example, a phosphodiesterase inhibitor and FDA approved compound, pentoxifylline, can systemically reduce inflammation and suppress synthesis of TNFα, preventing cisplatin nephrotoxicity in vivo [31]. Antioxidants such as Tempol [32], increased physical exercise [155], weight loss [156], and novel inhibitors such as Biochanin A [26] and Fasudil [47] have pleiotropic effects in addition to decreasing TNFα expression, with demonstrated in vivo renoprotective effects against cisplatin-induced AKI.

Cellular damage and its associated molecular products are thought to be key triggers for inflammation after acute tissue injury [157]. Activated and injured renal parenchymal cells secrete an array of chemokines promoting chemotaxis of acute inflammatory cell populations such as neutrophils, macrophages, and T cells [158,159]. Though not exhaustive, major chemokines secreted include CXCL1, CXCL8, CCL10, CCL2, CCL5, and IL-1β. As such, many chemokines serve as clinical biomarkers of AKI. The specific ranges of individual chemokines, such as CCL2, CCL5, IL-1β, IL-18, and IL-6, serving as sensitive and non-invasive markers for early detection of cisplatin-induced tubular injury have been discussed above.

In summary, TNFα is an acutely responsive cytokine downstream of NF-κB and SIRTs, capable of instigating a systemic network of inflammation, the bulk of which can exacerbate cisplatin-induced AKI. Inhibition of TNFα signaling suppresses the inflammatory cascade, ameliorating cisplatin-mediated renal injury.

#### 1.5.2. Inflammatory Cells

Chemokines facilitate the recruitment of leukocytes into sites of injury. Excess inflammatory cells can mediate additional renal dysfunction through release of cytokines, proteases, elastases, myeloperoxidases (MPO), and ROS. These substances can damage tissue directly, increase vascular permeability, and impair endothelial function. Adhesion molecules mediate leukocyte adhesion to other leukocytes, endothelial cells, and cell matrix, localizing leukocytes to specific sites of inflammation. Adhesion molecule, CD54^+^ (ICAM1), is expressed on vascular endothelium in response to TNFα stimulation. The junctional adhesion molecule C (JAM-C) has been reported to block the movement of neutrophils from inflamed tissue back into systemic circulation [160], a process referred to as reverse transendothelial migration [160]. JAM-C blocking antibodies have been shown to remove CD54^+^ neutrophils from cisplatin-induced inflamed tissues, mitigating the inflammatory response and ameliorating cisplatin-induced AKI [161]. Similarly, either genetic knockout [162], or pre-treatment of wildtype animals with CD54^+^ mAB significantly suppresses the initial neutrophil recruitment, diminishing MPO activity in whole kidney homogenates, reducing tubular injury, and protecting against cisplatin-induced elevated serum urea nitrogen and serum creatinine [163].

Lymphocytes educated in the thymus (T cells) make up approximately 30% of the circulating leukocytes in normal adults [164]. In cisplatin-induced AKI, activated CD4^+^ T cells rapidly and robustly infiltrate injured kidneys [165,166], mediating injury by producing cytokines such as TNFα [166]. In addition, activated CD4^+^ T cells express and shed death activator Fas ligand (FasL) and T cell immunoglobulin mucin (Tim-1, Hcvr1, Kim-1), mediating apoptosis (FasL) and receptor mediated phagocytosis (Kim-1) of injured renal tubular cells [52,167]. Genetic CD4 depletion studies, however, demonstrate mixed protection from cisplatin-induced AKI [166]. Furthermore, in tumour bearing mouse AKI models, CD4 depletion does not protect against cisplatin-induced AKI and results in worsening tumour burden [60]. One reason for this may be loss of CD4^+^ CD25^+^Foxp3 regulatory T cells (Tregs), which are known to protect the kidney during AKI [166]. Tregs can be induced from naïve T cells in the presence of TGFβ and IL-2, and in AKI, suppress pro-inflammatory responses by direct cell contact [168,169]. IL-2, a critical cytokine for the homeostasis of Tregs, upregulates IL-33 receptor (ST2). IL-233 is a novel hybrid cytokine of the two (IL-2+IL-33 [170]) Treatment of mice with IL-233 resulted in an expanded population of Tregs in lymphoid organs and renal compartments, and protection (decreased BUN, creatinine, and lower acute tubular necrosis) compared to vehicle treated cisplatin-induced AKI [170]. Additionally, Phospholipase A2 has been shown to increase IL-10 production and expand Treg populations in vivo and in vitro, ultimately providing protection against cisplatin-induced AKI [61].

Increased renal MPO, as discussed above, is associated with renal injury, which is produced by both neutrophils and macrophages (MΦ) [171]. Renal infiltration of MΦ (F4/80^Lo^CD11b^Hi^) and dendritic cells (DC, (F4/80^Hi^CD11b^Lo^)) can be identified 1–3 days after cisplatin-induced injury [172]. DC/MΦ depletion experiments exacerbate AKI kidney lesions induced by cisplatin, suggesting DC/MΦ mediate a protective role in renal injury [173]. Conversely, DC/MΦ depletion has been shown to decrease cisplatin-induced AKI severity or have no effect on cisplatin-induced AKI [174]. Conflicting data sets on the role of DC/MΦ in cisplatin-induced AKI demonstrate the need for additional studies to better define the role and function of DC/MΦ subtypes [172,175].

In summary, inflammation is a necessary and protective evolutionary response. However, excessive and prolonged stimulation of inflammatory cytokines and inflammatory cell mediators exacerbates renal injury. Cytokines released by leukocytes and injured renal tubular cells are instrumental in controlling the extent of inflammation. Tregs have demonstrated immunosuppressive roles in cisplatin-induced AKI. DC/MΦ subpopulations may effect protection against nephrotoxicity, but deeper and more carefully defined cell population studies are warranted.

### 1.6. Autophagy in Cisplatin-Induced AKI

Autophagy, the cellular process of self-catabolism, is a method by which cells break down and recycle cytoplasmic proteins and organelles in order to maintain intracellular homeostasis [15]. Stimulated by stress, injury, or nutrient deprivation, autophagy is essential for cellular metabolic maintenance and survival [176]. Autophagy is largely described in the literature as either elevated basal expression of autophagosome-bound microtubule-associated protein 1A/1B-light chain 3 phosphatidylethanolamine conjugate, (LC3-II), or a decrease in autophagosome substrate, p62. LC3-II is stimulated in both in vivo [177] and in vitro [178] models of cisplatin-induced epithelial injury. Further, the expression of autophagy indicator, LC3-II, after injury is largely considered a protective and self-defense mechanism [179].

HDAC6 is a major regulator of autophagosome maturation and autophagosome-lysosome fusion. Cisplatin-induced AKI stimulates HDAC6 expression and activity. HDAC6 inhibition results in increased expression of autophagy proteins (ATG7, Beclin-1) [64]. HDAC6 inhibition and resulting stimulation of autophagy, was associated with reduced renal oxidative stress, suppressed TNFα and IL-6 expression, inhibition of biomarkers NGAL and KIM1, and suppressed tubular cell apoptosis ultimately attenuating cisplatin-induced AKI [64]. Similarly, the plant-derived flavonoid, Scutellarin [180], administered prior to cisplatin, increased ATG5 and ATG7 expression while reducing LC3-II and p62 expression relative to cisplatin treatment alone, suggesting Scutellarin maintains or increases the rate of autophagosome clearance in spite of cisplatin-induced injury. While maintaining autophagy, Scutellarin also inhibits inflammation and apoptotic processes in vivo. The autophagy stimulation has also demonstrated protective effects in cisplatin-induced AKI. For example, treatment with metformin (effecting elevated LC3-II and phosphorylated AMPKα) results in suppressed tubular cell apoptosis, reduced inflammatory cell accumulation in the kidneys of treated animals, and attenuated cisplatin-induced AKI [42]. Additionally, two widely tested HDAC inhibitors, suberoylanilide hydroxamic acid (SAHA) and trichostatin A (TSA), protect the kidneys in cisplatin-induced AKI by enhancing autophagy [181]. Further, exogenous supplementation or enhanced expression of autophagy associated proteins, such as 14-3-3ζ and ATG16L, have also demonstrated alleviated cisplatin-induced AKI in vivo and in vitro by activating autophagy [179,182].

Inversely, pharmacological and genetic approaches to inhibit autophagy have decreased cell survival in response to cisplatin-mediated tubular injury. For example, tubular epithelial cells prepared from *Atg7*^−/−^ mice are more susceptible to cisplatin-induced caspase activation and apoptosis compared to *Atg7*^+/+^ cells exposed to cisplatin [183]. in vivo, administration of autophagy inhibitor, chloroquine, worsens cisplatin-induced AKI. Similarly, genetic disruption of either *Atg5* (*Atg5*^−/−^) [184] or Atg7 (*Atg7*^−/−^) [183] results in exacerbated apoptosis and enhanced AKI compared to *Atg5*^+/+^ or *Atg7*^+/+^ cisplatin administered mice. Lastly, protein kinase Cδ (PKCδ) and protein kinase c delta (PKCD) mediated suppression of autophagy has been recently shown to aggravate cisplatin nephrotoxicity, by promoting tubular cell death [185,186].

Lastly, mitophagy, a form of selective autophagy responsible for removing damaged or dysfunctional mitochondria, is mediated by the PINK1/Parkin pathway. Both PINK1 and Parkin are increased in kidney tissues after cisplatin-induced AKI in mice. PINK1 and Parkin knockout mice develop a more severe cisplatin-induced AKI, suggesting that PINK1/Parkin-mediated mitophagy is necessary in order to protect against cisplatin-induced AKI [187].

In summary, recent progress has been made in identifying the role of autophagy, and autophagy crosstalk with apoptosis and inflammation. The renoprotective benefit of autophagy stimulation potentiates the generation of multiple, unique targets, for therapeutic interventions in AKI. However, it should be noted that an increase in basal autophagosomes (increased basal LC3-II) can be attributed to either I) a decrease in lysosomal turnover of autophagosomes or II) the stimulation of autophagosome production. Studying autophagy in cisplatin-induced AKI requires careful controls, use of lysosomal inhibitors to study autophagic flux, and secondary comparisons of p62, for example, in order to ensure quality data interpretation is achieved.

### 1.7. Klotho in Cisplatin-Induced AKI

Klotho is a transmembrane protein expressed in multiple tissues and cell types. However, Klotho expression is particularly high in the kidney, specifically proximal [188] and distal convoluted tubules [189]. Recently, Klotho has garnered attention as the anti-aging protein [190]; functioning as a humoral factor with pleiotropic activities including regulation of oxidative stress [191], growth factor signaling [192], and ion homeostasis [193]. Further, studies have demonstrated secreted Klotho is also involved in organ protection. The intracellular form of Klotho can suppress inflammation-mediated cellular senescence [194] and mineral metabolism [193]. Tubular cell damage is a common consequence of cisplatin treatment [37]. In settings of acute renal injury, urinary Klotho levels are reduced below baseline in both humans [195] and in mouse models [37] of cisplatin-induced AKI. Moreover, reducing Klotho expression with genetically deficient Klotho (Kl^−/+^) mice prior to cisplatin treatment, exacerbates cisplatin-induced AKI. In summary, expression of Klotho is associated with organ protection, reduced oxidative stress, and regulation of growth factor signaling. Modulation of circulating and renal specific expression of Klotho may prove therapeutic and reduce AKI in patients treated with cisplatin [37].

### 1.8. Cisplatin-Induced Chronic Kidney Disease (CKD)

Cisplatin-induced renal injury is widely accepted as a model of AKI. Clinically, the need to understand the pathomechanism of AKI in order to intervene is paramount, as mortality after an AKI event is high (20–25% of hospital mortality). Additionally, in the intensive care unit, the incidence of AKI is 50–70% with greater than 50% mortality [196]. Lastly, even patients who survive AKI experience 28% mortality within the first year of their precipitating AKI episode [197]. Ultimately, those who survive their immediate AKI, still face long-term outcomes such as a two-fold [198] higher risk of developing chronic kidney disease (CKD) [199] and end-stage renal disease [198]. The pathomechanism of AKI to CKD transition is not yet well known [200], however, a growing area of research incorporates cisplatin in modeling AKI to CKD progression [30,60,98,201,202]. Though no consensus exists in the dosing or timing of a cisplatin-induced CKD models, different groups have used doses of 7–9 mg/kg/week consecutively for 4 weeks [98], 1 mg/kg twice weekly for 10 weeks [203], two injections of 15 mg/kg two weeks apart [202]. Ten mg/kg/week for 4 weeks [201] in both wildtype and CD4 T cell knockout results [60] in 100% mortality of recipient mice.

Cisplatin-induced CKD models faithfully recapitulate clinically relevant characteristics of CKD such as renal fibrosis, uremia, and biomarkers of CKD, such as progressive and chronically elevated plasma creatinine [204] and NGAL [205]. Both protein and mRNA levels of Klotho, the anti-aging protein discussed above, have been shown to be suppressed in cisplatin-induced CKD mice aged to 20 weeks. As Klotho functions in phosphate retention [206], loss of Klotho increases serum phosphate levels [206], accelerating aging and age-related cardiovascular and renal diseases in mice and humans [207]. To this point, Hu et al. demonstrated that high phosphate dietary consumption exacerbates AKI to CKD transition indicated by worsening renal fibrosis. Additionally, Klotho administration, during the AKI to CKD transition substantially ameliorated renal injury and fibrosis [208]. Additionally, Noiri et al. [201] demonstrated that oral administration of semicarbazide-sensitive amine oxidase inhibitor (PXS-4728A) successfully suppresses interstitial fibrosis and oxidative stress in their mouse model of cisplatin-induced CKD (10 mg/kg cisplatin, once weekly for 3 weeks.)

In summary, ongoing research addressing cisplatin-induced CKD will provide invaluable insight into therapeutics capable of treating both the acute and chronic repercussions of clinical cisplatin administration. A stronger understanding of cisplatin-induced CKD will ultimately increase the efficacy, and quality, of cisplatin treatment for patients.

#### Urinary Exosomes in Cisplatin-Induced AKI

Urinary exosomes are from every segment of the nephron, including podocytes [209]. Exosomes are 50–90 nm vesicles. An exosome is created inside the cell when a segment of the cell membrane invaginates and is endocytosed. The internalized segment is broken into smaller vesicles that can be expelled from the cell. The released vesicles are called exosomes. Exosomes are secreted by cells under normal and pathological conditions under control of RNA called “exosomal shuttle RNA”. The detection of urinary exosomal transcription factors may provide understanding of cellular regulatory pathways as well as being biomarkers of disease.

Exosomes were isolated by differential centrifugation and found to contain activating transcription factor 3 (ATF3) in rat models following acute injury at times earlier than the increase in serum creatinine [210]. An early decrease in release of aquaporin-2 in urinary extracellular vesicles was found after cisplatin treatment in rats [211]. Exosomal fetuin-A increased 52.5-fold at day 2 (1 day before serum creatinine increase and tubule damage) and remained elevated 51.5-fold at day 5 (peak renal injury) after cisplatin injection in rats [212]. Urinary excretion of exosomal organic anion transporter 5 (Oat5) increased in the rats treated with cisplatin and decreased when renal injury was ameliorated by N-acetylcysteine co-administration [213] Content of glutamyl aminopeptidase in microvesicular and exosomal fractions of urine is an early and predictive biomarker of renal dysfunction in cisplatin-induced AKI in rats [214]. Exosomes released by human umbilical cord mesenchymal stem cells protect against cisplatin-induced renal oxidative stress and apoptosis in rats in vivo and in vitro in rat kidney epithelial cells [215]. In summary, measurement of factors in urinary exosomes may offer insights into cellular regulatory pathways and early diagnosis of cisplatin-induced AKI.

## 2. Discussion

The proximal tubules of the kidney are the principal site of cisplatin toxicity. Cisplatin concentrates in the proximal tubules due to the localization and expression of cisplatin uptake transporters. The pathophysiology of AKI involves dysregulation of oxidative stress, apoptosis, necrosis, local and systemic inflammation, inflammatory mediators, and autophagy (Figure 1). Modulation of any single arm of the AKI system (inhibition of pro-inflammatory cytokines or oxidative stress, or stimulation of autophagy and immune suppressive cell populations or cytokines) results in amelioration of cisplatin-induced AKI. There are many areas of ongoing research in cisplatin-induced AKI, such as in the generation of more clinically relevant animal models, the characterization of autophagy, and further interrogation of the AKI to CKD transition. 

## Figures and Tables

**Figure 1 ijms-20-03011-f001:**
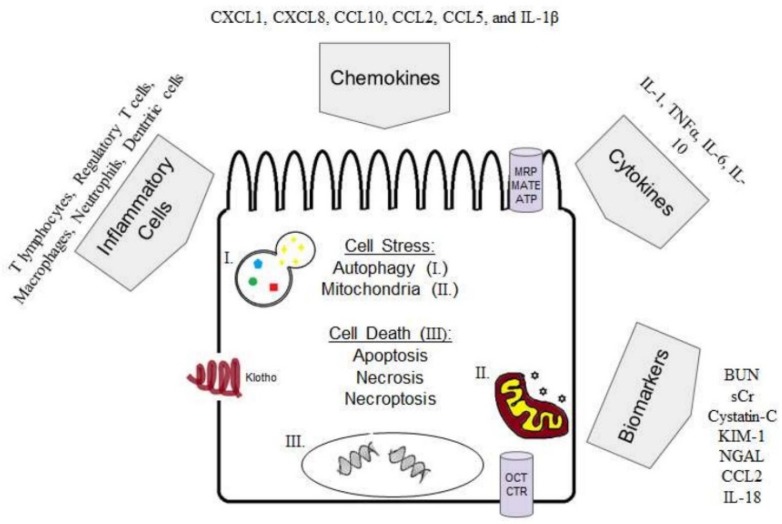
The complex pathophysiology of cisplatin-induced AKI. (I.) autophagosome/lysosome. (II.) mitochondria (III.) fragmented DNA (cell death). Abbreviations: organic cation transporters (OCTs), multidrug resistance-associated proteins (MRPs), multi-antimicrobial extrusion protein (MATEs), chemokine (C-X-C motif) ligand (CXCL), chemokine (C-C motif) ligand (CCL), interleukin-1 beta (IL-1β), tumour necrosis factor (TNF). blood urea nitrogen (BUN), serum creatinine (SCr) kidney injury molecule-1 (KIM-1), neutrophil gelatinase –associated lipocalin (NGAL), interleukin-18 (IL-18).

**Table 1 ijms-20-03011-t001:** Routine and novel biomarkers of cisplatin-induced AKI.

Type	Model					
	Low Dose, Short-Term		Low Dose, Long-Term		High Dose, Short-Term	
Cisplatin dosage	5–15 mg/kg		5–15 mg/kg		20–30 mg/kg	
Study duration	<7 Days		>7 Days		<7 Days	
Routine AKI biomarker	BUN [33] (40–60 mg/dL) SCr [34] (0.1–0.5 mg/dL)	** Cystatin-C [35] (1–1.3 µg/mL) KIM-1 [36] (10–12 fold increase) NGAL [37] (0.3–1.3 µg/mL) CCL2 [38] (0.5–10 fold increase mRNA)	BUN [39] (30–150 mg/dL) SCr [40] (0.5–3 mg/dL)	KIM-1 [41] (0.05–0.1 mg/mL) NGAL [41] (20–40 mg/mL)	BUN [42] (50–200 mg/dL) SCr [43] (0.5–3 mg/dL)	* Cystatin-C [44] (14–18 ng/mL) KIM-1 [45] 50–100 fold increased) NGAL/SCr [46] (6–10 µg/mg) CCL2 [46] (10–30 fold increase mRNA) IL-18 [47] (20–25 pg/mg)
Timing of detection	>48 h	>72 h	>48 h	>72 h	>48 h	>72 h
Novel AKI biomarker	miRNAs [5] (10 fold increase): miR-130a, 151–3p, 218, 320, 680, 138, 152, 221, 328, 685	Urinary Wnt4 [48] (0.1–0.7 µg/mg) Urinary ARL13B [49] (15–45 fold increase)	N/A	Renal IL-33 [50] (0.3–1.0 µg/mg)	N/A	Urinary NGAL [51] (40–120 units/g SCr) Urinary FasL [52] (45–150 pg/mL) Renal IL-33 [50] (0.3–1.0 µg/mg)
Timing of detection	<24 h	>24 h	N/A	>24 h	N/A	>24 h

Hours, hrs. Blood Urea Nitrogen, BUN. Kidney Injury Molecule-1, KIM1. Neutrophil Gelatinase-Associated Lipocalin, NGAL. Chemokine ligand 2, CCL2. Fas Ligand, FasL. Interleukin 33, IL-33. N-acetyl-β-d-glucosaminidase, NAG. Interleukin 18, IL-18. Creatinine, SCr. ADP Ribosylation Factor Like GTPase 13B, ARL13B. MicroRNA, MiRNA. * Mouse, ** Rat.

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
