# Peer review of "Recent Advances in Models, Mechanisms, Biomarkers, and Interventions in Cisplatin-Induced Acute Kidney Injury"

_ijms, 2019, doi:10.3390/ijms20123011_

Round 1

Reviewer 1 Report

The manuscript described a recent progress in the basic research on cisplatin nephrotoxicity with a strong advocate on urinary biomarkers.

1. A recent review on the similar topic is more extensive and through with nice figures : Volarevic V1, Djokovic B2, Jankovic MG3, Harrell CR4, Fellabaum C4, Djonov V5, Arsenijevic N2: Molecular mechanisms of cisplatin-induced nephrotoxicity: a balance on the knife edge between renoprotection and tumor toxicity. J Biomed Sci. 2019 Mar 13;26(1):25. doi: 10.1186/s12929-019-0518-9. The manuscript may be concentrated on the urinary markers. Hence, the tile could be modulated to include urinary biomarker to direct a new aspect of this disease especially miR. The discussion of urinary exosomes may need to be included.

2. The critical and negative view on the therapeutic approach on transporters seems to be one-sided. The authors may need to cite a positive following paper: Katsuda H1, Yamashita M, Katsura H, Yu J, Waki Y, Nagata N, Sai Y, Miyamoto K.: Protecting cisplatin-induced nephrotoxicity with cimetidine does not affect antitumor activity. Biol Pharm Bull. 2010;33(11):1867-71. In fact, ref. 24 was intended to manipulate transporters by doxycycline although much wider roles were identified.

3. The critique on the higher doses in murine models is also one-sided. As mice are resistant to nephrotoxic substances, especially B6/black strain, higher doses are necessary to produce clinical relevant cisplatin AKI models, which is my understanding.

4. A novel participant of klotho in cisplatin nephropathy is intriguing. However, ref. 183 dealt with chronic models, not AKI models. Please provide relevant papers.

5. Each title needs to be numbered. Some seem to be subtitles which should be sub-numbered.

6. Ref. 194 may be Kuro-o, M.

Author Response

Reviewer 1

1. A recent review on the similar topic is more extensive and through with nice figures : Volarevic V1, Djokovic B2, Jankovic MG3, Harrell CR4, Fellabaum C4, Djonov V5, Arsenijevic N2: Molecular mechanisms of cisplatin-induced nephrotoxicity: a balance on the knife edge between renoprotection and tumor toxicity. J Biomed Sci. 2019 Mar 13;26(1):25. doi: 10.1186/s12929-019-0518-9. The manuscript may be concentrated on the urinary markers. Hence, the tile could be modulated to include urinary biomarker to direct a new aspect of this disease especially miR. The discussion of urinary exosomes may need to be included.

Response: Title has been changed to: Recent advances in models, mechanisms, biomarkers and interventions in cisplatin-induced acute kidney injury. A discussion on urinary exosomes in cisplatin-induced AKI has been added (Line 560)

2. The critical and negative view on the therapeutic approach on transporters seems to be one-sided. The authors may need to cite a positive following paper: Katsuda H1, Yamashita M, Katsura H, Yu J, Waki Y, Nagata N, Sai Y, Miyamoto K.: Protecting cisplatin-induced nephrotoxicity with cimetidine does not affect antitumor activity. Biol Pharm Bull. 2010;33(11):1867-71. In fact, ref. 24 was intended to manipulate transporters by doxycycline although much wider roles were identified.

Response: This reference has now been discussed (line 64).

3. The critique on the higher doses in murine models is also one-sided. As mice are resistant to nephrotoxic substances, especially B6/black strain, higher doses are necessary to produce clinical relevant cisplatin AKI models, which is my understanding.

Response: We have expanded the discussion to be less one- sided (line 106)

4. A novel participant of klotho in cisplatin nephropathy is intriguing. However, ref. 183 dealt with chronic models, not AKI models. Please provide relevant papers.

Response: Ref 183 has been removed and ref 184 has been added

5. Each title needs to be numbered. Some seem to be subtitles which should be sub-numbered.

Response: Titles have been numbered as suggested.

6. Ref. 194 may be Kuro-o, M.

Response: Reference 194, now 196, has been corrected

Reviewer 2 Report

This comprehensive and clear review on Cisplatin AKI elegantly summarizes recent publications on new experimental models (rodents with tumors mimicking human counterpart ) and molecular pathways of injury especially in mitochondria .

This review will give useful and synthetic informations for clinicians and researchers.

It fits well with the goal of the journal.

Author Response

Reviewer 2

This comprehensive and clear review on Cisplatin AKI elegantly summarizes recent publications on new experimental models (rodents with tumors mimicking human counterpart ) and molecular pathways of injury especially in mitochondria .

This review will give useful and synthetic informations for clinicians and researchers.

It fits well with the goal of the journal.

Response: We thank the reviewer for these comments

Reviewer 3 Report

This manuscript reviewed the recent advances in the molecular pathophysiology of cisplatin-induced acute kidney injury(AKI). The review is organized and well written. If some figures can be added to demonstrate the complex mechanisms of cisplatin-induced AKI, the manuscript will be more comprehensive.

Author Response

Reviewer 3

This manuscript reviewed the recent advances in the molecular pathophysiology of cisplatin-induced acute kidney injury(AKI). The review is organized and well written. If some figures can be added to demonstrate the complex mechanisms of cisplatin-induced AKI, the manuscript will be more comprehensive.

Response: Table 1 shows routine and novel biomarkers of cisplatin-induced AKI.

A new figure (Figure 1) demonstrating the complex mechanisms of cisplatin-induced AKI has been added

Round 2

Reviewer 1 Report

The manuscript has been improved.